# A Fully Analog Pipeline for Portfolio Optimization

**James S. Cummins**     **Natalia G. Berloff**
Department of Applied Mathematics and Theoretical Physics
University of Cambridge
Wilberforce Road, Cambridge CB3 0WA, UK
`jsc95@cam.ac.uk`

## Abstract

Portfolio optimization is a ubiquitous problem in financial mathematics that relies on accurate estimates of covariance matrices for asset returns. However, estimates of pairwise covariance are notoriously poor and calculating time-sensitive optimal portfolios is energy-intensive for digital computers. We present an energy-efficient, fast, and fully analog pipeline for solving portfolio optimization problems that overcomes these limitations. The analog paradigm leverages the fundamental principles of physics to recover accurate optimal portfolios in a two-step process. Firstly, we utilize equilibrium propagation, an analog alternative to backpropagation, to train linear autoencoder neural networks to calculate low-rank covariance matrices. Then, analog continuous Hopfield networks output the minimum variance portfolio for a given desired expected return. The entire efficient frontier may then be recovered, and an optimal portfolio selected based on risk appetite.

## 1   Introduction

Portfolio optimization involves creating an investment portfolio that balances risk and return. The objective is to allocate assets optimally to maximize expected returns while minimizing risk. Naturally, this problem is of great interest to financial organizations and is pivotal in risk management. However, the problem, formulated by Markowitz's mean-variance model [1], suffers from problems in practice. Namely, it is well known that estimates of pairwise covariance between assets are notoriously poor [2]. Data samples tend to include significant amounts of noise, distorting the underlying relationships between assets. To overcome this issue, factor models were introduced that vastly reduce the dimensionality [3]. Factor methods produce low-rank covariance matrices that retain only the largest eigenvalues and discard small eigenvalues associated with noise. Despite this development, the computation of optimal portfolios remains energy-intensive as the efficient frontier is mapped out in return-variance space. In high-frequency trading, this becomes a time-sensitive computation as assets are purchased and sold on microsecond timescales [4], and portfolios must be regularly rebalanced so as not to exceed risk appetites. Much attention has been focused on portfolio optimization in the high-frequency domain [5, 6, 7], including the use of evolutionary algorithms to update efficient frontiers [8]. While algorithmic enhancements provide incremental gains, the exploration of alternative hardware paradigms has the potential to drive significant advancements. By using fundamental principles such as minimizing entropy, energy, and dissipation [9], or, perhaps, incorporating quantum phenomena like superposition and entanglement [10], we can advance and surpass classical computations of these problems. At the forefront of this drive to alternate architectures is the integration of analog, physics-based algorithms and hardware, which involve translating complex optimization problems into universal spin Hamiltonians [11, 12, 13]. Indeed, the mean-variance portfolio optimization framework can be encoded into a Hamiltonian's coupling strengths with the physical system recovering the Hamiltonian's ground state, which corresponds to the optimal portfolio solution [14, 15]. Efficient mapping from the original problem description to spin Hamiltonian enables the problem to remain manageable despite increasing complexity [16].

38th Second Workshop on Machine Learning with New Compute Paradigms at NeurIPS 2024(MLNCP 2024).

## 2   Mean-variance optimization

We define $\mu_i$ as the expected return of asset $i$, and $[\mathbf{\Sigma}]_{ij} = \sigma_{ij} = \mathrm{Cov}(i,j)$ as the covariance between assets $i$ and $j$. The decision variables are $w_i$, the proportion of the total investment in asset $i$. For a universe of securities with $n$ assets, the Markowitz mean-variance portfolio optimization problem is

$$
\begin{aligned}
\min_{\mathbf{w}} \quad & \mathbf{w}^{\mathrm{T}}\mathbf{\Sigma}\mathbf{w} \\
\text{s.t.} \quad & \boldsymbol{\mu}^{\mathrm{T}}\mathbf{w} = R, \\
& \mathbf{1}^{\mathrm{T}}\mathbf{w} = 1, \\
& 0 \leq w_i \leq 1,
\end{aligned}
\tag{1}
$$

for $i = 1, \ldots, n$, and the condition $w_i \geq 0$ prohibits shorting [14]. The variance $\mathbf{w}^{\mathrm{T}}\mathbf{\Sigma}\mathbf{w}$ quantifies the portfolio risk for positive semidefinite matrix $\mathbf{\Sigma}$, while $R$ is the desired expected return of the portfolio. $\boldsymbol{\mu}$ and $\mathbf{\Sigma}$ are not known a priori and must be estimated from historical data. The efficient frontier is calculated by solving (1) for various $R$. The efficient frontier is the set of portfolios that minimize the risk for a given $R$. We illustrate a frontier in Appendix A for a toy model with $n = 2$ assets. It was recently suggested that portfolio optimization problems could be solved on analog spatial-photonic Ising machines for equal-weighted portfolios, that is, $w_i \in \{0, 1/q\}$ for $q$ selected assets [15]. We go beyond this constraint by utilizing analog Hopfield networks and consider the quadratic continuous optimization problem (1). In Section 3, we aim to recover the optimal asset weights $\mathbf{w}$, given known expected returns $\boldsymbol{\mu}$ and covariance matrix $\mathbf{\Sigma}$.

## 3   Continuous Hopfield network

A continuous Hopfield network is a type of Hopfield neural network which has continuous states and dynamics [17]. It is an analog computational network for solving optimization problems. For a network of size $n$, the $i$-th network element at time $t$ is described by a real input $x_i(t)$, and the network dynamics are governed by

$$
\frac{\mathrm{d}x_i}{\mathrm{d}t} = -p(t)x_i + \sum_{j=1}^{n} J_{ij}v_j + m_i,
\tag{2}
$$

where $v_i = g(x_i)$ is a nonlinear activation function, $p(t)$ is an annealing parameter, $m_i$ are the offset biases, and $J_{ij}$ are elements of the symmetric coupling matrix $\mathbf{J}$. Should $g(x)$ be a non-decreasing function, then the steady states of the continuous Hopfield network (2) are the minima of the Lyapunov function

$$
E = p(t) \sum_{i=1}^{n} \int_{0}^{v_i} g^{-1}(x)\mathrm{d}x - \frac{1}{2}\sum_{i,j=1}^{n} J_{ij}v_iv_j - \sum_{i=1}^{n} m_iv_i,
\tag{3}
$$

We choose the functional form of $g(x)$, such that when $p(t) \to 0$, the minima of $E$ occur for $v_i \in [0,1]$ and correspond to the minima of $-\mathbf{v}^{\mathrm{T}}\mathbf{J}\mathbf{v}$. Therefore, by setting $\mathbf{J} = -\mathbf{\Sigma}$, we can minimize the variance $\mathbf{w}^{\mathrm{T}}\mathbf{\Sigma}\mathbf{w}$ of problem (1). To satisfy the constraints in problem (1) we introduce Lagrange multiplier-like scalars $\lambda_1, \lambda_2$ and seek to minimize the expression $H = \mathbf{w}^{\mathrm{T}}\mathbf{\Sigma}\mathbf{w} + \lambda_1(\boldsymbol{\mu}^{\mathrm{T}}\mathbf{w} - R)^2 + \lambda_2(\mathbf{1}^{\mathrm{T}}\mathbf{w} - 1)^2$. Therefore, after discarding constants, we seek to minimize

$$
H = -\frac{1}{2}\mathbf{w}^{\mathrm{T}}\mathbf{J}\mathbf{w} - \mathbf{m}^{\mathrm{T}}\mathbf{w},
\tag{4}
$$

where $\mathbf{J} = -2\mathbf{\Sigma} - 2\lambda_1\boldsymbol{\mu}\boldsymbol{\mu}^{\mathrm{T}} - 2\lambda_2\mathbf{1}\mathbf{1}^{\mathrm{T}}$, and $\mathbf{m} = 2R\lambda_1\boldsymbol{\mu} + 2\lambda_2\mathbf{1}$. Equation (4) can be directly encoded into the Hopfield network (2), and if required, $\mathbf{m}$ can be absorbed into $\mathbf{J}$ by introducing an additional auxiliary spin. The non-decreasing monotonic function $g(x)$ is chosen to be the logistic function $g(x) = 1/[1 - \exp(-x)]$ to limit possible values of $v_i$ such that $0 \leq v_i \leq 1$. We illustrate the Hopfield network dynamics in Appendix B for a randomly generated covariance matrix $\mathbf{\Sigma}$ and expected return vector $\boldsymbol{\mu}$. The energy minimization properties of Hopfield networks make them particularly suitable for solving combinatorial optimization problems. Further extensions have been proposed to increase convergence to optimal states in challenging optimization problems. For example, the first-order Eq. (2) can be momentum-enhanced and replaced with a second-order equation leading to Microsoft's analog iterative machine [18] or Toshiba's bifurcation machine [19].

## 4 Low-rank approximation

We now focus on calculating a low-rank approximation of the covariance matrix, which will be used in (1). If $\mathbf{x}_i \in \mathbb{R}^n$ are the $i$-th sample of asset returns over $N$ total samples, and we assume that $\mathbb{E}[\mathbf{x}] = \mathbf{0}$, then the sample covariance matrix is $\mathbf{S} = \frac{1}{N} \sum_{i=1}^{N} \mathbf{x}_i \mathbf{x}_i^{\mathrm{T}}$. When the number of samples $N$ is of the same magnitude as $n$, then the sample covariance matrix usually suffers a large estimation error [2, 20]. Many low-rank factor analysis techniques exist to improve the covariance matrix estimate. Here, we consider asset returns $\mathbf{x}$ as random variables that follow the model

$$\mathbf{x} = \mathbf{As} + \mathbf{e}, \tag{5}$$

where $\mathbf{x} \in \mathbb{R}^n$ is the observed data, $\mathbf{A} \in \mathbb{R}^{n \times r}$ is a factor loading matrix, $\mathbf{s} \in \mathbb{R}^r$ is the vector of latent variables, and $\mathbf{e} \in \mathbb{R}^n$ is uncorrelated random noise, where $r \ll n$ [21]. Here, $\mathbf{s}$ represents macroeconomic factors like the growth rate of the GDP, unemployment, inflation etc. Under the assumption that $\mathbf{s}$ and $\mathbf{e}$ are uncorrelated, the covariance matrix is then $\mathbf{\Sigma} = \mathbb{E}[\mathbf{x}\mathbf{x}^{\mathrm{T}}]$. This gives

$$\mathbf{\Sigma} = \mathbf{A}\mathbb{E}[\mathbf{s}\mathbf{s}^{\mathrm{T}}]\mathbf{A}^{\mathrm{T}} + \mathbb{E}[\mathbf{e}\mathbf{e}^{\mathrm{T}}] \tag{6}$$

$$= \mathbf{A}\mathbf{P}\mathbf{A}^{\mathrm{T}} + \mathbf{\Psi}, \tag{7}$$

where $\mathbf{P} \equiv \mathbb{E}[\mathbf{s}\mathbf{s}^{\mathrm{T}}] \in \mathbb{R}^{r \times r}$ has $\mathrm{rank}(\mathbf{P}) \leq r$, $\mathrm{rank}(\mathbf{A}) \leq r$, and $\mathbf{\Psi}$ is a diagonal matrix containing the variance of noise on its diagonal. Since $\mathrm{rank}(\mathbf{AB}) \leq \min(\mathrm{rank}(\mathbf{A}), \mathrm{rank}(\mathbf{B}))$, then $\mathrm{rank}(\mathbf{A}\mathbf{P}\mathbf{A}^{\mathrm{T}}) \leq r$. Therefore, we have decomposed the covariance matrix $\mathbf{\Sigma}$ into a positive semidefinite low-rank matrix plus a positive semidefinite diagonal matrix. Defining $\mathbf{M} \equiv \mathbf{A}\mathbf{P}\mathbf{A}^{\mathrm{T}}$, low-rank factor analysis concerns the estimation of $\mathbf{M}$ and $\mathbf{\Psi}$. To calculate $\mathbf{M}$ and $\mathbf{\Psi}$ we solve the minimization problem

$$\begin{aligned} \min_{\mathbf{M}, \mathbf{\Psi}} \quad & ||\mathbf{S} - \mathbf{M} - \mathbf{\Psi}||_{\mathrm{F}}^2 \\ \text{s.t.} \quad & \mathrm{rank}(\mathbf{M}) \leq r, \\ & \mathbf{\Sigma} \succeq 0, \end{aligned} \tag{8}$$

where $|| \cdot ||_{\mathrm{F}}$ denotes the Frobenius norm [22]. We present a common digital computing method in Appendix (C) for solving problem (8) based on principal component analysis (PCA). The eigendecomposition in PCA becomes computationally expensive as the data size grows. Alternatively, autoencoders – particularly when implemented using stochastic gradient descent – can handle larger datasets and higher-dimensional data more efficiently than PCA [23]. Additionally, when integrating dimensionality reduction as part of a larger neural network framework, an autoencoder can be easily embedded within the pipeline, whereas PCA would need to be applied as a separate pre-processing step [24].

## 5 Linear autoencoders

A linear autoencoder is a classic neural network model for unsupervised learning that is trained to learn the identity function. The input and output layers have the same number of nodes, while the middle layer has fewer nodes. It aims to approximate the input through learning linear encodings and decodings between input and latent space. The encoder $\mathbf{B} \in \mathbb{R}^{r \times n}$ maps input $\mathbf{X} = [\mathbf{x}_1, \ldots, \mathbf{x}_N] \in \mathbb{R}^{n \times N}$ into a low-dimensional latent space $[\mathbf{s}_1, \ldots, \mathbf{s}_N]$, and the decoder $\mathbf{A} \in \mathbb{R}^{n \times r}$ maps $[\mathbf{s}_1, \ldots, \mathbf{s}_N]$ back to the original representation $\mathbf{X}$. We therefore recover the same model as in Eq. (5), and training the linear autoencoder becomes the minimization problem [25]

$$\min_{\mathbf{A}, \mathbf{B}} \quad ||\mathbf{X} - \mathbf{A}\mathbf{B}\mathbf{X}||_{\mathrm{F}}^2. \tag{9}$$

We do not explicitly express the learnable biases in the network as these may be absorbed into the encoder $\mathbf{B}$ and decoder $\mathbf{A}$ by introducing an auxiliary row into $\mathbf{X}$ that is permanently clamped to values of 1. We illustrate the training of a linear autoencoder in Fig. (1)(a)-(c) with the backpropagation method and compare it to PCA in Fig. (1)(h). A linear autoencoder is related to PCA. Indeed, under mild nondegeneracy conditions, any $\mathbf{A}$ at a local minimizer recovers the top rank-$r$ eigenspace of $\mathbf{X}\mathbf{X}^{\mathrm{T}}$ [26]. However, unlike actual PCA, the coordinates of the output of the middle layer in the network are correlated and are not sorted in descending order of variance [27]. Autoencoder neural networks typically use backpropagation to train the weights. However, backpropagation is energy-intensive and not biologically plausible.

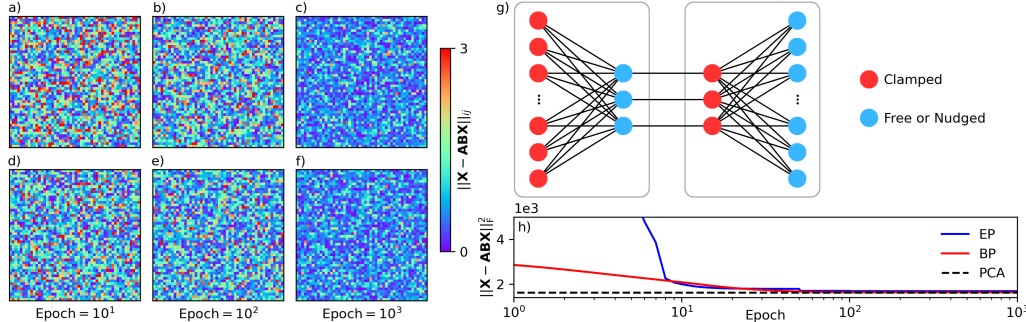

Figure 1: Training a linear autoencoder via (a)-(c) backpropagation (BP), and (d)-(f) with equilibrium propagation (EP). Input and output layers have size 50, while the single hidden layer has size 5. The networks are trained on 50 vectors $\mathbf{x}_1, \ldots, \mathbf{x}_{50}$ of size 50 whose elements are randomly sampled from the normal distribution $N(0, 1)$. (a)-(f) illustrate the element-wise absolute difference between the $50 \times 50$ matrix $\mathbf{X} = [\mathbf{x}_1, \ldots, \mathbf{x}_{50}]$ and its reconstructed output $\mathbf{ABX}$ at different epochs. (g) An illustrative example of the encoder/decoder Hopfield network structure trained with EP. (h) The overall network loss for BP and EP over epoch time. The black horizontal dashed line corresponds to the loss of the equivalent PCA method described in Appendix C.

## 6 Equilibrium propagation

On dedicated analog hardware, equilibrium propagation is an energy-efficient alternative to backpropagation [28]. Therefore, in the supervised learning setting studied here, it may be used to train the weights of a linear autoencoder. Equilibrium propagation is an energy-based model because it relies on the concept of energy minimization to learn and make predictions. We consider the continuous Lyapunov function (3) with $p(t) = 1$ and $m_i = 0$ for all $i$, where here the symmetric coupling weights $J_{ij}$ are to be learned, and nonlinear activation function $g(x)$ need not be the same as in Section 3. Neurons $x_i$ are split in three sets: the input neurons, which are always clamped, the hidden neurons, and the output neurons. The discrepancy between the desired output $\mathbf{y}$ and the realized output $\hat{\mathbf{x}}$ is measured by the cost function

$$C = \frac{1}{2}||\mathbf{y} - \hat{\mathbf{x}}||_2^2, \tag{10}$$

which forms part of the total energy function $F = E + \beta C$. The clamping factor $\beta \geq 0$ is a real-valued scalar that allows the output neurons to be weakly clamped [29]. The continuous-time dynamical system evolves according to the differential equation of motion

$$\frac{\mathrm{d}x_i}{\mathrm{d}t} = -\frac{\partial F}{\partial x_i} = -\frac{\partial E}{\partial x_i} - \beta \frac{\partial C}{\partial x_i}, \tag{11}$$

which is formed of two parts. The first is the internal force induced by the internal Hopfield energy, given by Eq. (2) for all $i$, and the second, the external force, is induced by the cost function $C$ as

$$-\beta \frac{\partial C}{\partial x_i} = \beta(y_i - x_i), \quad i \in \mathcal{Y}, \tag{12}$$

for nodes in the output layer $\mathcal{Y}$. Equilibrium propagation has two modes: the free phase and the weakly clamped phase. In the free phase $\beta = 0$ and only the inputs are clamped. The network then converges to a fixed point $\mathbf{x}^*$ and the output units are read out. In the weakly clamped phase $\beta > 0$, which induces an external force that acts on the output units as in Eq. (12). This force nudges the outputs from their fixed point values in the direction of the target values $y_i$. This perturbation propagates among the hidden neurons before a new fixed point $\mathbf{x}_\beta^*$ is found. Then, another weakly clamped phase is executed, this time with $\beta \to -\beta$ leading to the weakly clamped equilibrium $\mathbf{x}_{-\beta}^*$. It was shown that the weakly clamped phase implements the propagation of error derivatives with respect to the synaptic weights [29]. In the limit $\beta \to 0$, the update rule is

$$\Delta J_{ij} \propto \frac{1}{\beta}\left(\left.\frac{\partial F}{\partial J_{ij}}\right|_{\mathbf{x}_\beta^*} - \left.\frac{\partial F}{\partial J_{ij}}\right|_{\mathbf{x}_{-\beta}^*}\right), \tag{13}$$

which is a second-order approximation to the standard backpropagation derivative [30]. The process is iterated, at each step updating the weights $J_{ij}$ to minimize the loss function $C$. We choose activation function

$$g(x) = \begin{cases} x & \text{if } |x| \leq c \\ c \cdot \text{sgn}(x) & \text{otherwise,} \end{cases} \tag{14}$$

with constant $c$, so that under the condition that $|x_i| \leq c$ for all $i$, Eq. (2) is linear and can thus represent a linear autoencoder. In this case, the output $\hat{\mathbf{x}}$ of Eq. (2) is then the solution to the linear differential equation $d\mathbf{x}/dt = (\mathbf{J} - \mathbf{I})\mathbf{x}$, and therefore

$$\hat{\mathbf{x}} = \lim_{t \to \infty} \mathbf{x}(t) = \lim_{t \to \infty} \exp\{(\mathbf{J} - \mathbf{I})t\}\mathbf{x}(0). \tag{15}$$

The constant $c$ in Eq. (14) is chosen to be large enough such that after training, all neurons obey $|x_i| \leq c$, and we can associate the Hopfield network as a linear autoencoder. To achieve a steady state in Eq. (15), at least one eigenvalue of $\mathbf{J} - \mathbf{I}$ should be zero, with all others having a negative real part.

**Proposition**. *We state, with proof given in Ref. [26], that for any fixed $n \times r$ matrix $\mathbf{A}$, Eq. (9) attains its minimum for $\mathbf{B} = (\mathbf{A}^{\mathrm{T}}\mathbf{A})^{-1}\mathbf{A}^{\mathrm{T}}$.*

**Lemma**. *The $n \times n$ matrix $\mathbf{J} - \mathbf{I}$, where $\mathbf{J} = \mathbf{AB}$, has at least one zero eigenvalue, with all others having negative real part.*

*Proof.* $\mathbf{J} = \mathbf{AB} = \mathbf{A}(\mathbf{A}^{\mathrm{T}}\mathbf{A})^{-1}\mathbf{A}^{\mathrm{T}}$, and therefore

$$\mathbf{J}^2 = \mathbf{A}(\mathbf{A}^{\mathrm{T}}\mathbf{A})^{-1}\mathbf{A}^{\mathrm{T}}\mathbf{A}(\mathbf{A}^{\mathrm{T}}\mathbf{A})^{-1}\mathbf{A}^{\mathrm{T}} \tag{16}$$

$$= \mathbf{A}(\mathbf{A}^{\mathrm{T}}\mathbf{A})^{-1}\mathbf{A}^{\mathrm{T}}, \tag{17}$$

which shows that $\mathbf{J}$ is idempotent, that is $\mathbf{J}^2 = \mathbf{J}$. It follows that $\mathbf{J}$ is a projection operator on the column space $C(\mathbf{J})$ along its null space $N(\mathbf{J})$. The $n$ eigenvalues $\lambda_i$ of $\mathbf{J}$ are either 0 or 1: $\lambda_i \mathbf{x}_i = \mathbf{J}\mathbf{x}_i = \mathbf{J}^2\mathbf{x}_i = \lambda_i \mathbf{J}\mathbf{x}_i = \lambda_i^2 \mathbf{x}_i$, which implies $\lambda_i \in \{0, 1\}$. By construction, $\mathbf{J}$ has rank at most $r$, and therefore there are at least $n - r$ zero eigenvalues. It follows that there are between 1 and $r$ nonzero eigenvalues of $\mathbf{J}$, which must have value $\lambda_i = 1$. Since $\mathbf{J} - \mathbf{I}$ has eigenvalues $\mu_i = \lambda_i - 1$, then $\mu_i \in \{-1, 0\}$. Therefore, $\mathbf{J} - \mathbf{I}$ has between 1 and $r$ zero eigenvalues, with all others being equal to $-1$.

The Lemma guarantees that should equilibrium propagation learn the weights that minimize Eq. (9), the corresponding Hopfield network will converge to a steady state. Yet, during training, this will, in general, not be the case, and positive eigenvalues of $\mathbf{J} - \mathbf{I}$ will produce exponential growth in Eq. (15). However, Eq. (15) only holds in the linear regime of the activation function (14). Exponential growth is prevented by the symmetric clipping incorporated into the nonlinear activation function $g(x)$ for neurons with $|x_i| > c$.

In the linear regime, the overall network dynamics is represented by the square matrix $\lim_{t \to \infty} \exp\{(\mathbf{J} - \mathbf{I})t\}$, which for linear autoencoders we seek to decompose into its non-square constituent parts: encoder $\mathbf{B}$ and decoder $\mathbf{A}$. We achieve this by treating the encoder and decoder as separate Hopfield networks, as shown in Fig. (1)(g), each with their own energy function. The encoder settles into an equilibrium representing the latent vector $\mathbf{s}$ without taking into account the decoder. $\mathbf{s}$ is then used as a fixed input to the decoder which then settles into its own equilibrium. The decoder then undergoes the weakly clamped phases, and its weights are updated according to Eq. (13). The encoder weights also need to be optimized to lower the reconstruction loss at the decoder output, which is achieved by setting

$$\frac{\partial C}{\partial x_i} = \lim_{\beta \to 0} \frac{1}{2\beta} \left( \left. \frac{\partial F}{\partial x_i} \right|_{\mathbf{x}_\beta^{(\text{dec})}} - \left. \frac{\partial F}{\partial x_i} \right|_{\mathbf{x}_{-\beta}^{(\text{dec})}} \right), \quad i \in \mathcal{Y}, \tag{18}$$

in Eq. (11), where $\mathbf{x}_\beta^{(\text{dec})}$ is the weakly clamped decoder equilibrium state, and Eq. (18) only pertains to neurons in the encoder output layer $\mathcal{Y}$. Equation (18) follows from the fact that it can be shown that equilibrium propagation also allows for finding the gradient of the loss with respect to the input [31]. We note that $\mathbf{J}$, which contains the couplings of the continuous Hopfield network, is now a $(n + r) \times (n + r)$ matrix on account of the number of nodes in the encoder and decoder networks. Nonetheless, the factor loading matrix $\mathbf{A}$ can be recovered as the $n \times r$ block corresponding to the nodes of the decoder output layer. The equilibrium propagation training procedure is illustrated in Fig. (1)(d)-(f) and compared to backpropagation and PCA in Fig. (1)(h).

# 7 Results

We collect real data samples $\mathbf{x}_i \in \mathbb{R}^n$ from stock returns of a selection of $n = 100$ stocks in the S&P 500 index. We restrict ourselves to only $N = 100$ observations such that the sample covariance matrix has a tendency to contain significant noise. Two continuous Hopfield networks, structured as the encoder and decoder parts of a linear autoencoder, are trained using equilibrium propagation. The latent variables $[\mathbf{s}_1, \ldots, \mathbf{s}_N]$ are calculated as the subset $\mathcal{Y}$ of steady-state solutions of the encoder network, while the factor loading matrix $\mathbf{A}$ is the $n \times r$ block of the decoder matrix representation $\lim_{t \to \infty} \exp\{(\mathbf{J} - \mathbf{I})t\}$ corresponding to its output layer $\mathcal{Y}$. In practice, we cannot take the limit to infinity, and instead, we use a suitably large value of $t$ such that $\exp\{(\mathbf{J} - \mathbf{I})t\}$ changes minimally from $t$ to $t + 1$. We depict the full-rank sample covariance matrix and the equilibrium propagation-based low-rank approximation in Figs. (2)(a) and (b) respectively. Figure (2)(c) then illustrates the element-wise absolute difference between these two covariance matrices. The low-rank approximation is plugged into (1) and solved for the portfolio weights $\mathbf{w}$ using the continuous Hopfield network of Eq. (2). We minimize the portfolio variance subject to the constraint $\boldsymbol{\mu}^{\mathrm{T}}\mathbf{w} = R$ for incremental values of $R$. In Fig. (2)(d), we plot the corresponding variances and returns for range $R = [0, 1]$. The efficient frontier is identified, and an optimal portfolio can be selected based on risk appetite.

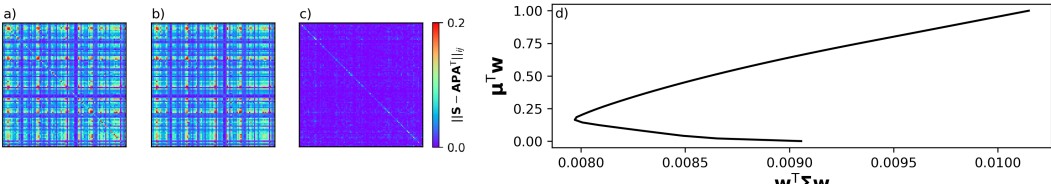

Figure 2: (a) The sample covariance matrix $\mathbf{S}$ for $n = 100$ financial stocks selected from the S&P 500 index using $N = 100$ time series samples. (b) The $r = 10$ low-rank approximation $\mathbf{APA}^{\mathrm{T}}$ of the covariance matrix, as calculated by training a continuous Hopfield network via equilibrium propagation. (c) The element-wise absolute difference between the sample covariance matrix and its low-rank approximation. (d) The hyperbola in variance-return space for possible optimal portfolios. Each point along the hyperbola is calculated by solving (1) for a specific return value $R$.

Analog Hopfield networks can be implemented as electronic circuits [32] and photonic neural networks [33]. Photonic systems operate on picosecond to femtosecond timescales as high bandwidth signals flow through a single optical waveguide. Consequently, such implementations can have dense connectivity while maintaining fast convergence times. However, physical analog platforms are subject to noise sensitivity, thermal effects, and non-idealities in circuit components which can degrade performance. In addition, real-world portfolio optimization problems often involve complex constraints such as transaction costs, market liquidity, regulatory requirements, and cardinality constraints. While some of these can be readily incorporated into the objective function (4), for example, an $\ell^1$-norm can enforce sparsity to satisfy a cardinality constraint, others take more complex forms. To address the limitations, a hybrid approach that combines analog Hopfield networks with digital computing could be explored.

# 8 Conclusions

This paper introduces a fully analog pipeline for portfolio optimization problems. Starting with raw data samples, the proposed pipeline leverages the energy-efficient analog operation of continuous Hopfield networks to calculate optimal portfolio weights. The analog pipeline distinguishes itself from traditional digital methods by its speed and scalability, with applications in time-sensitive domains such as high-frequency trading. At the heart of the pipeline are continuous Hopfield networks, used in two separate applications: autoencoder neural networks and minimum variance portfolios. By shifting to analog architectures, we reduce the reliance on binary logic operations typical of digital systems, paving the way for a more energy-efficient approach to computation. This efficiency can reduce power consumption in data centers and other computing environments, addressing the growing energy demands of digital computing. Specifically, companies can reduce their energy consumption while optimizing large portfolios as part of their risk management processes.

## Acknowledgments and Disclosure of Funding

J.S.C. acknowledges the PhD support from the EPSRC. N.G.B. acknowledges support from HORIZON EIC-2022-PATHFINDERCHALLENGES-01 HEISINGBERG Project 101114978 and Weizmann-UK Make Connection Grant 142568.

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

# A  Efficient frontier

The efficient frontier represents the set of optimal portfolios that offer the highest expected return for a given level of risk, or the lowest risk for a given return. It is derived by plotting the risk-return profiles of various portfolios, with the frontier itself being the curve where no portfolio exists with both a higher return and lower risk. For the toy model in Fig. (3) with $n = 2$ assets, the entire set of possible portfolios lies on a single hyperbola.

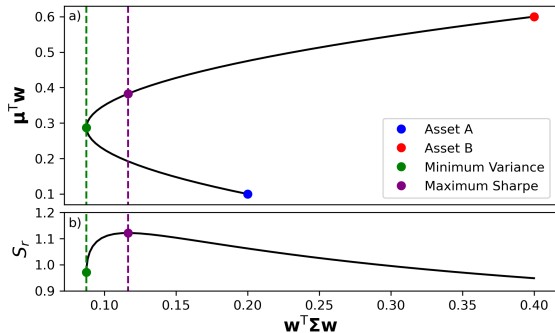

Figure 3: (a) The hyperbola in variance-return space for a portfolio of $n = 2$ assets $A$ and $B$. The positively sloped portion of this hyperbola is the efficient frontier. The expected returns are $\mu_A = 0.1$ and $\mu_B = 0.6$. The (co)variances are $\sigma_{AA} = 0.2$, $\sigma_{BB} = 0.4$, and $\sigma_{AB} = \sigma_{BA} = -0.1$. The blue circle represents the portfolio consisting only of asset $A$, and the corresponding investment weights are $\mathbf{w} = [1, 0]^{\mathrm{T}}$. Likewise, the red circle is the portfolio consisting only of asset $B$. The minimum variance portfolio, shown as a green circle, is the combination of weights $\mathbf{w}$ that minimizes the total variance $\mathbf{w}^{\mathrm{T}}\boldsymbol{\Sigma}\mathbf{w}$. The purple circle is the portfolio that maximizes the Sharpe ratio $S_r$. The Sharpe ratio is a measure of risk-adjusted return and is defined as $S_r = \boldsymbol{\mu}^{\mathrm{T}}\mathbf{w}/\sqrt{\mathbf{w}^T\boldsymbol{\Sigma}\mathbf{w}}$. (b) The Sharpe ratio $S_r$ for each portfolio in the efficient frontier. We now see that the purple circle is indeed the portfolio that maximizes the Sharpe ratio.

# B  Hopfield network dynamics

The quadratic continuous optimization problem (4) is solved using continuous Hopfield network (2). The trajectories of the neurons $x_i(t)$ are illustrated in Fig. (4) along with the value of objective function (4) in time for a portfolio of $n = 25$ assets.

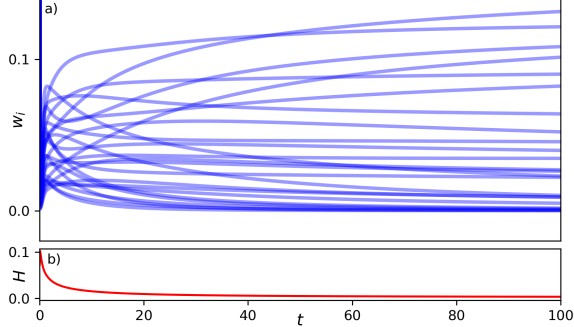

Figure 4: (a) Hopfield network dynamics for a portfolio of $n = 25$ assets with $R = \lambda_1 = \lambda_2 = 1$. The dynamical system evolves according to Eq. (2), which in turn minimizes Eq. (4). Each line represents one asset weight $w_i$. (b) The value of expression (4) during the network dynamics. Covariance matrix $\boldsymbol{\Sigma}$ and expected return vector $\boldsymbol{\mu}$ are calculated from sampling $N = 10$ observations of returns $\mathbf{x}$ from IID random normal variables $x_j \sim N(1, 1)$, where $j = 1, 2, \ldots, N$. The low number of observations $N$ results in a noisy positive semidefinite covariance matrix $\boldsymbol{\Sigma}$ whose pairwise entries $\sigma_{ij}$ are nonzero. The externally controlled annealed parameter has form $p(t) = p_0(1 - t/T)$, where $T$ is the total annealing period. Here, $p_0 = 0.01$ and $T = 100$.

# C Low-rank method

A common classical procedure for calculating matrix $\mathbf{M}$ in problem (8) using digital computers is given by the following steps:

1. Construct the singular value decomposition (SVD) of $\mathbf{S}$. Since $\mathbf{S}$ is symmetric, its eigende-composition is the same as the SVD, and we write $\mathbf{S} = \mathbf{U}\boldsymbol{\Lambda}\mathbf{U}^{\mathrm{T}}$, where $\mathbf{U}$ is the matrix of eigenvectors and $\boldsymbol{\Lambda}$ is the diagonal matrix of eigenvalues.

2. Derive from $\boldsymbol{\Lambda}$ the matrix $\boldsymbol{\Lambda}_r$ formed by replacing with zeros the $n - r$ smallest eigenvalues on the diagonal of $\boldsymbol{\Lambda}$.

3. Compute and output $\mathbf{M} = \mathbf{U}\boldsymbol{\Lambda}_r\mathbf{U}^{\mathrm{T}}$ as the rank-$r$ approximation to $\mathbf{S}$.

Under the assumption $\mathbb{E}[\mathbf{x}] = \mathbf{0}$, the SVD method exactly replicates PCA. The rank of $\mathbf{M}$ is at most $r$: this follows from the fact that $\boldsymbol{\Lambda}_r$ has at most $r$ non-zero values. Indeed, the Eckart-Young-Mirsky theorem proves that this procedure yields the matrix of rank less than or equal to $r$ with the lowest possible Frobenius error [34]. The diagonal matrix is estimated as $\boldsymbol{\Psi} = \mathrm{diag}(\mathbf{S} - \mathbf{M})$, where $\mathrm{diag}(\cdot)$ represents a diagonal matrix whose elements are $[\boldsymbol{\Psi}]_{ii} = [\mathbf{S} - \mathbf{M}]_{ii}$ and $[\boldsymbol{\Psi}]_{ij} = 0$ for $i \neq j$ [35]. In addition, we constrain $[\boldsymbol{\Psi}]_{ii} \geq 0$, since the diagonal elements correspond to variances of the error variables. This guarantees that $\boldsymbol{\Sigma}$ is positive semidefinite.

