# OpenReview forum: "A Fully Analog Pipeline for Portfolio Optimization"
_NeurIPS.cc/2024/Workshop/MLNCP — MLNCP Poster_

### Official Review · Reviewer_x8yb · 2024-09-23
**An interesting use case of Equilibrium Propagation, however very difficult to read**

**Rating:** 5
**Confidence:** 5

**Review:**

**Summary.** The paper proposes to apply Equilibrium Propagation (EP), an algorithm suited for analog hardware, to portfolio optimization. Namely, given $n$ assets $X_k$ and the covariance matrix $\Sigma:=\mathbb{E}[X \cdot X^\top]$ associated to these assets and $w_i$ the proportion of asset $i$ inside the portfolio, the optimal asset allocation should minimize $E:=w^\top \cdot \Sigma \cdot w$ under positivity and convexity constraint on $w$ and ensuring a return $R$ (assuming some value for the assets given by another vector $\mu$). The subtlety though is that the empirical covariance matrix $\Sigma$ may not be used itself, but a *low-rank* approximation thereof. This can be done by using a *linear* autoencoder $A\cdot B$ where $A$ and $B$ have to be learned such that $\ell := || X - A \cdot B \cdot X ||_F^2$ is minimized. Therefore, the goal is to find $A, B$ which minimize $\ell$ *under the constraint that* the energy $E$ is minimized. The approach is demonstrated on 100 observations of 100 stocks taken from the S\&P 500 index.

**Strengths**. It is an interesting use case for analog computing that is worth spreading at the workshop.

**Weakenesses**.

- I still have a hard time understanding well how the problem is framed how the application of EP reads here. Eq. 1 is very misleading because it wrongly conveys that  $E:=w^\top \cdot \Sigma \cdot w$ is the outer objective, but it actually turns out (if I understood correctly?) to be the *inner* objective, i.e. the constraint to be satisfied *while minimizing the reconstruction error* $\ell := || X - A \cdot B \cdot X ||_F^2$. There are lots of different notations that are introduced for the free variables, for the various parameters such that I really fail to understand eventually what are precisely the variables of the *outer* optimization problem ($\ell$ as defined above) and **how the energy function $E$ is precisely defined in terms of the variables appearing inside** $\ell$.

- "it can be shown that equilibrium propagation also allows for finding the gradient of the loss with respect to the input" should be attributed to [1], which precedes [28] by two years.

[1] Scellier, B. (2021). A deep learning theory for neural networks grounded in physics. arXiv preprint arXiv:2103.09985
[28] Tom Van Der Meersch, Johannes Deleu, and Thomas Demeester. Training a hopfield variational autoencoder with equilibrium propagation. In Associative Memory & Hopfield Networks in 2023, 2023.

---

### Official Review · Reviewer_mmAW · 2024-10-03
**A use-case of continuous Hopfield networks and equilibrium propagation for portfolio optimization**

**Rating:** 7
**Confidence:** 4

**Review:**

This paper is about solving the Markowitz mean-variance portfolio optimization problem using algorithms compatible (in principle) with analog hardware, specifically: continuous Hopfield networks, and equilibrium propagation.

The portfolio optimization problem is formulated as a constrained optimization problem: minimizing the risk for a given level of expected returns. The challenge is that the covariance matrix and expected returns are not explicitly known ; these must be estimated using samples from historical data. The authors estimate the covariance matrix using a low-rank approximation method, which is known to yield better results than naive estimation methods. Specifically, they use an autoencoder, consisting of a linear encoder and a linear decoder. The encoder and decoder take the form of Continuous Hopfield Networks, trained using Equilibrium Propagation.

Once the encoder and decoder are trained, a new continuous Hopfield network is built, representing the low-rank approximation of the covariance matrix, and whose ground state is an approximate solution of the original portfolio optimization problem.



**Presentation**

In Section 2, it could be useful to remind the reader that one of the challenges of Problem (1) is that the vector of expected returns mu and the covariance matrix Sigma are not known. We must estimate their values using samples from historical data.

The analytical expression of J in Eq.3 is (re)defined several times: line 60, line 64, and again line 151, which makes it a little difficult to follow.
I am wondering if Section 3 (Continuous Hopfield Networks) would be better placed between Section 5 (Linear Autoencoders) and Section 6 (Equilibrium Propagation), in which case you might consider removing the first two definitions of J (line 60 and line 64).
Alternatively, it could be clarified that Sections 2 and 3 are about “finding the optimal w given Sigma and mu”, whereas Sections 4, 5, 6 are about “finding good approximates of Sigma and mu”. (see also my third question below)



**Questions**

The original constrained optimization problem (Eq 1) is approximated by an unconstrained optimization problem using a soft penalty on the constraints (Eq 4). How does this affect the solution?

Similarly, in the unconstrained optimization problem, outputs are normalized during postprocessing. It is not directly clear if the v (asset allocation) obtained this way is a solution to the original problem. How does this method affect the solution?

If I understood correctly, there are not 2, but 3 different continuous Hopfield networks: one for the encoder, one for the decoder, and a third one whose coupling strengths correspond to the covariance matrix of the original portfolio optimization problem (which is used for inference, once the encoder and decoder have been trained). Is this correct?



**Minor remarks**

For the function E (Eq.3) to be a Lyapunov function of the dynamics (Eq.2), I believe that the nonlinear activation function g must be increasing. It is mentioned at line 66 that g is non-decreasing, but it could be explained earlier (at line 56) that this is a requirement for the E to be a Lyapunov function.

If I understood correctly, the method presented in Appendix C is a common method to solve the problem on a digital computer, but is not applicable on analog systems. If so, this could be clarified.

Ref [17] should probably be replaced by:
Hopfield, John J. "Neurons with graded response have collective computational properties like those of two-state neurons." Proceedings of the national academy of sciences 81.10 (1984): 3088-3092.

---

### Decision · Program_Chairs · 2024-10-10

Accept (Poster)